# Efficient Black-Box Planning Using Macro-Actions with Focused Effects

**Cameron Allen,**[1,2*] **Michael Katz,**[2] **Tim Klinger,**[2]
**George Konidaris,**[1] **Matthew Riemer,**[2] **Gerald Tesauro**[2]

[1]Brown University
[2]IBM Research

## Abstract

The difficulty of deterministic planning increases exponentially with search-tree depth. Black-box planning presents an even greater challenge, since planners must operate without an explicit model of the domain. Heuristics can make search more efficient, but goal-aware heuristics for black-box planning usually rely on goal counting, which is often quite uninformative. In this work, we show how to overcome this limitation by discovering macro-actions that make the goal-count heuristic more accurate. Our approach searches for macro-actions with focused effects (i.e. macros that modify only a small number of state variables), which align well with the assumptions made by the goal-count heuristic. Focused macros dramatically improve black-box planning efficiency across a wide range of planning domains, sometimes beating even state-of-the-art planners with access to a full domain model.

## 1 Introduction

In classical planning, an agent must select a sequence of deterministic, durationless actions to transition from a known initial state to a state that satisfies the desired goal condition. Planning assumes the agent has access to a model of the effects of its actions, which it uses to reason about potential plans. Usually this model takes the form of a PDDL description or finite-domain representation (Fox and Long 2003; Helmert 2009), which specifies the preconditions and effects of each action. However, in black-box planning (Lipovetzky, Ramirez, and Geffner 2015; Jinnai and Fukunaga 2017), the model is instead defined implicitly by a simulator that the agent can query to generate state transitions.

In general, planning is hard: determining whether a plan exists to reach the goal is *PSPACE*-complete (Bylander 1994). Heuristic search eases this computational burden by guiding the search towards promising solutions. Of course, heuristic search is only useful with a good heuristic. In classical planning, much work has gone into the development of domain-independent methods that automatically construct heuristics to exploit as much problem structure as possible from the formal PDDL problem description (Bonet and Geffner 2001; Hoffmann and Nebel 2001; Helmert 2006; Helmert

---

*The first author was at IBM Research for part of this work. Please send any correspondence to <csal@cs.brown.edu>. Code repository and supplementary materials for this paper are available at https://github.com/camall3n/focused-macros.

and Domshlak 2009; Helmert et al. 2014; Pommerening et al. 2015; Keyder, Hoffmann, and Haslum 2014; Domshlak, Hoffmann, and Katz 2015). However, black-box planners have no formal domain description to exploit, and are therefore limited to less-informed heuristics.

One simple, domain-independent heuristic that is compatible with simulator-based planners is the goal-count heuristic (Fikes and Nilsson 1970), which counts the number of state variables that differ between a given state and the goal. The two basic assumptions of the goal-count heuristic are: a factored state space (i.e. there are state variables to count), and a known goal condition (i.e. there is a reason to modify variables). A third, more subtle assumption is that the problem can be decomposed into subproblems, where each state variable can be treated as an approximately independent subgoal. Unfortunately, this subgoal independence assumption is invalid for most planning problems of practical interest, and thus the goal-count heuristic is often misleading.

A second domain-independent strategy for improving planning efficiency is to use abstraction in the form of high-level macro-actions. When macro-actions are added to the set of low-level actions, they can reduce search tree depth at the expense of increasing the branching factor. In some cases, this has been shown to improve planning efficiency, particularly when the macro-actions cause the problem's subgoals to become independent (Korf 1985). We further explore this idea in the context of black-box planning by constructing macros that are well aligned with the goal-count heuristic.

We begin by examining why goal counting becomes uninformative for certain sets of actions. We show that both goal-count accuracy and planning efficiency are linked to how many state variables actions can modify at once. Our investigation suggests a compelling strategy for improving the usefulness of the goal-count heuristic: learning *focused* macro-actions that modify as few variables as possible, so as to align with the assumptions made by the goal-count heuristic. This approach also seems well-aligned with human problem solving, for example, among expert Rubik's cube solvers, where focused macros are essential for the most efficient planning strategies.

We describe a method for discovering focused macro-actions and test it on several classical planning benchmarks, restricting our attention to quickly finding feasible plans, rather than optimal ones, with the goal of minimizing the

number of simulator queries. Our learned macro-actions enable reliable and efficient planning, making dramatically fewer calls to the simulator and improving solve rate on most domains. Our approach is designed to improve the goal-count heuristic, but it is compatible with more sophisticated black-box planning techniques as well—with similar improvement. In some cases, black-box planning with focused macros is even competitive with approaches that have access to much more detailed problem information.

# 2 Background

We consider the problem of black-box planning (Jinnai and Fukunaga 2017), where the planning agent does not have access to a declarative action description. Formally, we define a black-box planning domain using the following quantities:

- A set of states $S$, where each state is represented as a vector $v$ and each element $v_i$ is a state variable assignment from some finite domain $D(v_i)$;

- An action applicability function $A(s)$ that outputs the set of valid grounded actions for the given state $s \in S$;

- A deterministic[1] simulator function $\text{Sim}(s, a)$, which the agent can query to determine the next state $s'$ after executing the action $a \in A(s)$ from state $s \in S$.

Each of the above quantities is fixed for all planning problem instances in the domain. A particular problem instance additionally contains:

- A start state, $s_0 \in S$;

- A goal condition $G$, represented as a list of variable assignments to all (or some subset) of the state variables.

The planner's objective is to find a plan that connects state $s_0$ to any state $s_G$ that satisfies $G$ via a sequence of actions. In general, actions can have associated costs, and an optimal plan is one that minimizes the sum of its action costs. Here we are concerned with planning efficiency, so we focus on *satisficing* solutions—that is, finding a plan as quickly as possible, regardless of cost. We measure planning efficiency in terms of the number of simulator queries (equivalently, the number of generated states) before finding a plan.

## 2.1 The Goal-Count Heuristic

The goal-count heuristic, $\#g$, is defined in terms of the problem-specific goal, $G$. For any goal condition $G$, $\#g(s)$ counts the number of variables in state $s$ whose values differ from those specified in $G$, with $\#g(s) = 0$ if and only if $s$ satisfies $G$. A well-known downside of the goal-count heuristic is its dependence on the size of $G$. In the extreme case where $|G| = 1$, the goal-count heuristic only separates goal states from non-goal ones. Nevertheless, due to the relative lack of information in black-box planning, the goal-count is often the only goal-aware heuristic available. Other planners may add additional components, such as state novelty (Francès et al. 2017), but the black-box versions of those planners still rely on goal counting at their core.

## 2.2 Macro-Actions

A *macro-action* (or *macro*), is a deterministic sequence of actions,[2] typically for the purpose of accomplishing some useful subgoal. To avoid confusion, we often refer to the original non-macro actions as "primitive" actions. Macros have parameters, preconditions, and effects, just like primitive actions, but in black-box planning, we again assume that the planner does not have access to such a declarative description. Instead, when planning with macro-actions, there are two alternatives. Either the action applicability function $A(s)$ and simulator function $\text{Sim}(s, a)$ are updated to additionally compute *macro-action* validity and effects in a single step, or alternatively, each primitive action in the macro must be simulated sequentially, with longer macros requiring more simulator queries.

# 3 Effect Size and Goal-Count Accuracy

The goal-count heuristic implicitly treats each state variable as an independent subgoal. There are two ways to satisfy this assumption exactly. The first is if each subgoal can be achieved in one step without modifying any other state variable. The second, more general way, explored by Korf [1985], is if each subgoal can be achieved in one step (possibly modifying other state variables) and the subgoals are serializable—i.e. there is an ordering of the state variables that retains previously-solved subgoals when solving new ones.

In general, an action can of course change many state variables, and the problem representation may not allow the subgoals to be serialized—both of which can cause the goal-count heuristic to be uninformative. However, for a heuristic to be useful, it does not need to be perfect; it simply needs to be *rank correlated* with the distance to the goal: higher true distances should correspond to higher heuristic values (Wilt and Ruml 2015). When the heuristic is perfectly rank correlated, there is a monotonic relationship between heuristic and true cost, and best-first search will always expand nodes in order of their true distance from the goal.

We hypothesize that if each action modifies only a small number of state variables, the problem will better match the assumptions of the goal-count heuristic, and thus the heuristic and true goal distance will be more positively rank correlated. We informally say such actions have "focused" effects, and we formalize this idea with the following definitions:

**Definition 1.** The *effect size of an action* is the maximum number of state variables whose values change by executing the action, over all states where the action is applicable.

**Definition 2.** The *effect size of a macro-action* is the maximum number of state variables, measured at the end of macro-action execution, that are different from their starting values, over all states where the macro-action is applicable, even if additional variables were modified during execution.

If our hypothesis above is correct, we expect the goal-count heuristic to be more accurate for domains where actions

---

[1]In general, black-box planning can include probabilistic effects, but we leave this more general case for future work.

[2]For simulators with probabilistic effects, macro-actions could in principle be generalized to more complex abstract skills incorporating state information, but that extension is beyond the scope of this work.

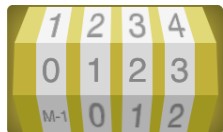

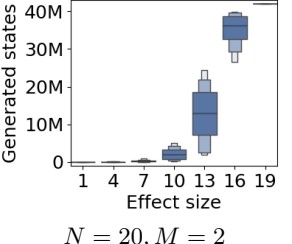
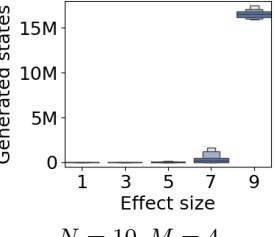

$N = 20, M = 2$  $N = 10, M = 4$

Figure 1: (Top) The Suitcase Lock domain with $N = 4$; (Bottom) Generated states vs. effect size for Suitcase Lock.

| Effect Size | N=10, M=2 | | N=5, M=4 | |
|---|---|---|---|---|
| $\bar{k}$ | $\rho_P$ | $\rho_S$ | $\rho_P$ | $\rho_S$ |
| 1 | 1.000 | 1.000 | 0.775 | 0.760 |
| 2 | 0.200 | 0.179 | 0.263 | 0.226 |
| 3 | 0.110 | 0.092 | 0.046 | 0.018 |
| 4 | 0.060 | 0.041 | 0.000 | -0.044 |
| 5 | 0.020 | 0.013 | – | – |
| 6 | 0.000 | -0.007 | – | – |
| 7 | 0.000 | 0.001 | – | – |
| 8 | 0.000 | -0.001 | – | – |
| 9 | 0.000 | 0.005 | – | – |

Table 1: Correlation results between the goal-count heuristic and true distance for Suitcase Lock. Actions with smaller effect size ($\bar{k}$) lead to significantly higher Pearson's correlation ($\rho_P$) and Spearman's rank correlation ($\rho_S$) coefficients.

have smaller effect size, and we further expect this to lead to an improvement in planning efficiency. In the following experiment, we see better rank correlation between heuristic and true distance for domains whose actions have low average effect size, and we see that this leads to an approximately exponential improvement in planning efficiency.

## 3.1 The Suitcase Lock Domain

To study the relationship between effect size and planning efficiency, we introduce the Suitcase Lock domain. The Suitcase Lock is a planning problem whose solution requires entering a combination on a lock with $N$ dials, each with $M$ digits, and $2N$ actions, half of which increment a deterministic subset of the dials (modulo $M$) and the other half of which decrement the same dials (see Figure 1, top). For each problem instance, a start state, goal state, and (fixed) action set are generated randomly, and a parameter $\bar{k}$ controls the mean effect size across all actions. This allows us to examine action effect size while holding other problem variables constant. For implementation details, see Appendix A of the supplementary materials.

**Focused Actions Improve the Goal-Count Heuristic** We first investigate the accuracy of the goal-count heuristic for two small Suitcase Lock problems. For each possible effect size, we compute the true distance between all pairs of states, and compare the results with the goal-count, treating the second state of each pair as the goal. We compute the average heuristic value for each true distance, and then compute the Pearson correlation and Spearman rank correlation coefficients between heuristic and distance. The results are shown in Table 1, where we see that actions with more focused effects (i.e. lower $\bar{k}$) lead to significantly higher correlation.

**Focused Actions Improve Planning Efficiency** Next, we run two planning experiments using the goal-count heuristic and greedy best-first search (GBFS). To evaluate planning efficiency, we measure the number of generated states needed to solve each instance, since we care about feasibile plans, rather than optimal ones.

Figure 1 (bottom) shows an approximately exponential relationship between effect size and planning time. When $M = 2$ and $k = 1$, the goal-count heuristic is exactly equal to the cost, and GBFS can generate at most $N^2$ states before finding the goal. By contrast, when $k = (N-1)$ the heuristic is maximally uninformative, and GBFS may generate $N \cdot 2^N$ states in the worst case, since it cannot expand any nodes with heurstic value $k$ until it has expanded all nodes with heuristic value $< k$. This exponential trend appears to hold even when state variables are not binary.

These results on the Suitcase Lock domain suggest that reducing effect size is a viable strategy for improving planning efficiency. To further investigate this idea, we propose a method for learning macro-actions with low effect size.

## 4 Learning Macros with Focused Effects

We search for macro-actions using best-first search (BFS) with a simulation budget of $B_M$ state transitions. We start the search at a randomly generated state, and the search heuristic is macro-action effect size—or infinity if the macro-action modifies zero variables—plus the number of primitive actions in the macro. Technically, we would need to evaluate each macro from every valid state to determine its effect size, which is clearly infeasible. In practice, we simply measure each macro's effect size once and assume it doesn't change (although we could easily relax this assumption by running the macro from multiple states).

We save the $N_M$ macro-actions with the lowest effect size, and ignore duplicate macro-actions that have the same net effect. To encourage diversity of macros, we can optionally repeat the search $R_M$ times, each time generating a new random starting state in which none of the existing saved macro-actions are valid, or until we fail to find such a starting state.[3] This ensures that we still find macros that apply in most situations, even if there are constraining preconditions.

The pseudocode for this procedure is in Algorithm 1. We

---

[3] Finding such a state may be as hard as planning, unless the simulator can be reset to generate new starting states; however, in practice, a random walk is often sufficient (see appx. Note B.2.1).

**Algorithm 1** Learn macro-actions with focused effects

---

**Input**: Starting state $s_0$, number of macro-actions $N_M$, number of repetitions $R_M$, search budget $B_M$
**Output**: List of macro-actions $L_M$

1: **Define** $g(m) := \text{length}(m)$
2: **Define** $h(s) := \begin{cases} |\text{net\_effects}(s - s_0)| & \text{if } > 0, \\ \infty & \text{otherwise} \end{cases}$
3: **Define** $f(s, m) := g(m) + h(s)$
   where $m$ is the macro (i.e. action sequence) from $s_0$ to $s$

4: Let $L_M$ be an empty list of macro-actions
5: Let $Q$ be a (max) priority queue of size $N_M/R_M$
6: **for** repetition $r$ in $\{1, ..., R_M\}$ **do**
7:   Run best-first search (BFS) from $s_0$ with budget $B_M/R_M$, minimizing heuristic $f(s, m)$
8:   **for** each state $s_i$ and macro $m_i$ visited by BFS **do**
9:     Store $m_i$ in $Q$, with priority $h(s_i)$
       // When $Q$ becomes full, the action sequences
       // with largest $h$-score will get evicted first
10:   **end for**
11:   Add each unique macro in $Q$ to $L_M$
12:   Clear $Q$
13:   $s_0 \leftarrow$ new random state, such that none of the macros in $L_M$ can run
14:   **if** $s_0$ is None **then**
15:     **break**
16:   **end if**
17: **end for**
18: **return** $L_M$

---

use $m_i$ to denote the macro whose action sequence generated state $s_i$. Consider an example with two primitive actions $a_1$ and $a_2$, where BFS starts at state $s_0$. Expanding $s_0$, the action $a_1$ generates $s_1$, and $a_2$ generates $s_2$. Expanding $s_1$, $a_1$ generates $s_3$, and $a_2$ generates $s_4$. Thus macro $m_4$, corresponding to state $s_4$, would be the action-sequence $[a_1, a_2]$, and we would evaluate its net effect by comparing $s_4$ with $s_0$.

# 5 Experiments

We evaluate our method by learning macro-actions in a variety of black-box planning domains and subsequently using them for planning. We use PDDLGym (Silver and Chitnis 2020) to automatically construct black-box simulators from classical PDDL planning problems. Additionally, we use two domain-specific simulators (for 15-puzzle and Rubik's cube) that have a different state representation to show the generality of our approach. See the appendix for implementation details and a discussion of how we selected the various macro-learning hyperparameters (sections B and E, respectively).

We select the domains to give a representative picture of how the method performs on various types of planning problems. For PDDLGym compatibility reasons, we restrict the domains to those requiring only `strips` and `typing`. For the domain-specific simulators, we select 15-puzzle and Rubik's cube in particular, because they present opposing challenges for our macro-learning approach. In 15-puzzle, primitive actions have very focused effects (each modifies

only the blank space and one numbered tile), but naively chosen macro-actions tend to have much larger effect sizes, and both primitive actions and macros have state-dependent preconditions. In Rubik's cube, actions and macros have no preconditions, but primitive actions are highly non-focused (each modifies 20 of the simulator's 48 state variables) and the state space is so large ($\sim 4.3 \times 10^{19}$ unique states (Rokicki 2014)) that black-box planning is unable to solve the problem efficiently.

## 5.1 Methodology

For each planning domain, we generate 100 problem instances with unique random starting states and a fixed goal condition.[4] All problem instances share the same state space, and the planner has access to the simulator function, the action applicability function, a vector of state information, and the goal condition. We emphasize that although the PDDLGym domains are specified using PDDL, the planner never sees the PDDL during either macro search or planning.

We learn focused macro-actions as described in Sec. 4 and add them to the set of primitive actions, which ensures that the same set of states can still be reached. These macros are then used to update the simulator and action applicability function, allowing the learned macros to execute in a single step for improved computational efficiency.[5] Note that updating the simulator in this way does not reduce search effort, only time. Even if the primitive actions in a macro were simulated one-by-one, the intermediate states are neither stored nor explored, and hence do not count towards the number of generated states.

The macros are learned once, for the first problem instance, and then reused on all remaining problem instances for that domain. In general, it can be challenging to incorporate macros into any planning algorithm, since one must weigh their search benefits against the increased branching factor. For simplicity, our experiments fixed the number of macros $N_M$ (see Table 2), but in principle $N_M$ could be chosen automatically based on which macros reduce the problem's average effect size.

To solve each planning problem, we use greedy best-first search (GBFS) with the goal-count heuristic and compare performance with the additional learned macro-actions versus with primitive actions alone. We measure planning efficiency as the number of simulator queries that the planner makes before finding a plan. This choice of performance metric is the most natural fit for black-box planning, and it allows for fair comparisons of algorithms across different implementation languages and hardware configurations.

In Table 2, we show the average solve rate and number of generated states (i.e. simulator queries) for each domain. Since we only pay the macro-learning cost $B_M$ for the first problem instance, we can in principle amortize this cost over the total number of problem instances. (Note that the $B_M$ values reported in the table are non-amortized and are separate from the number of generated states.) Except in the case of

---

[4] All problem instances are included in the code repository.

[5] See Appendix C for details on how we update the simulator and action applicability function to incorporate the learned macros.

| Domain | $N_M$ | $B_M$ | GBFS(A) | | GBFS(A+M) | | BFWS(A) | | BFWS(A+M) | | LAMA(A) | |
|---|---|---|---|---|---|---|---|---|---|---|---|---|
| | | | Gen | Sol | Gen | Sol | Gen | Sol | Gen | Sol | Gen | Sol |
| Depot | 8 | 50K | 58275.9 | **0.74** | **55132.4** | 0.60 | 75966.9 | **0.48** | **72205.8** | 0.34 | 46620.9 | 1.00 |
| Doors | 8 | 5K | 3050.7 | 1.00 | **512.6** | 1.00 | 4660.9 | 1.00 | **3057.3** | 1.00 | 293.0 | 1.00 |
| Ferry | 8 | 5K | 1875.8 | 1.00 | **1151.4** | 1.00 | 1209.9 | 1.00 | **1163.5** | 1.00 | 699.8 | 1.00 |
| Gripper | 8 | 5K | 7314.8 | 1.00 | **6277.0** | 1.00 | 44945.9 | 1.00 | **6295.9** | 1.00 | 6493.1 | 1.00 |
| Hanoi | 8 | 100K | 78433.6 | 0.78 | **6358.8** | **1.00** | 63455.2 | 1.00 | **3365.9** | 1.00 | 65496.4 | 1.00 |
| Miconic | 8 | 5K | 7559.4 | 1.00 | **1907.1** | 1.00 | 10269.2 | 1.00 | **1884.3** | 1.00 | 1316.7 | 1.00 |
| 15-Puz. | 192 | 32K | 30840.5 | 1.00 | **4952.4** | 1.00 | 109425.2 | 1.00 | **6290.1** | 1.00 | – | – |
| Rubik's | 576 | 1M | >2M | 0.00 | **171.3K** | **1.00** | >2M | 0.00 | **163.8K** | **1.00** | 9.13M | 1.00 |

Table 2: Black-box planning results for PDDLGym-based simulators (top), and domain-specific simulators (bottom). (A) - primitive actions only; (A+M) - primitive actions + focused macros; $N_M$ - number of macros; $B_M$ - macro-learning budget; Gen - generated states; Sol - solve rate; (bold) - best performance of each planner. The efficiency of both GBFS and BFWS($R_G^*$) are improved by adding focused macros. Note that LAMA is an informed planner with access to much more information than black-box planners, and is only included for reference.

Depot, we see that planning with focused macros increases solve rate and improves planning efficiency by up to an order of magnitude versus planning with primitive actions alone. In Rubik's cube, focused macros still perform better, even if we account for the *entire* macro-learning budget.

## 5.2 Comparisons with Other Planners

In addition to greedy best-first search (GBFS) with the goal-count heuristic, we also evaluate our method in conjunction with Best-First Width Search, or BFWS (Lipovetzky and Geffner 2017), a family of search algorithms that augment their search heuristic with a novelty metric computed using Iterated Width (IW) search (Lipovetzky and Geffner 2012).

We specifically use the best-performing black-box planning version of BFWS: BFWS($R_G^*$) (Francès et al. 2017). This version starts by running IW up to two times, with increasing precision, to generate a set $R_G^*$ of goal-relevant atoms. During search, each state $s$ is evaluated based on how many relevant atoms were satisfied at some point along the path to $s$. This forms a relevance count $\#r(s)$, which is combined with the goal-count $\#g(s)$ to compute the novelty width metric $w_{\#r,\#g}$. The algorithm runs GBFS using heuristic $(w, \#g, c)$, evaluating nodes first by width, breaking ties with $\#g$, and then breaking further ties with $c$, the cost to reach the node.

We ran BFWS on each domain and measured its planning efficiency (see Table 2). We followed Lipovetzky and Geffner [2017] and limited the width precision to $w \in \{1, >1\}$ on Depot and Rubik's cube to save computational resources.

Again we find that focused macros substantially improve planning efficiency, likely because the heuristic still uses goal counting at its core. Surprisingly, we found that BFWS did not perform significantly better than the primitive-action GBFS baseline. In fact, comparing against GBFS, we observe that focused macros alone are more beneficial for planning than the more sophisticated novelty-based heuristic.

As a point of reference, we also compared against LAMA (Richter and Westphal 2010) which has full access to a declarative representation of the problem—information far beyond

what is available to black-box planners. We ran the first iteration of LAMA on the same problems we used with PDDLGym, as well as a SAS$^+$ representation of the Rubik's cube, adapted from Büchner [2018]. On a different PDDL version of Rubik's cube, LAMA failed to complete the translation step before running out of memory (16GB). We find our method is competitive with LAMA, across the majority of domains, despite the fact that LAMA has access to more information. On the 100 hardest Rubik's cube problems from Büchner [2018], which neither primitive-action baseline can solve, LAMA generates 9.1 million states on average, whereas our approach generates only 171 thousand.

## 5.3 Comparison with Random Macros

One might wonder whether the improvements in planning efficiency are due to the macros' focused effects, or simply the fact that we are using macros at all. To isolate the source of the improvement, we conducted a second experiment using 15-puzzle and Rubik's cube. Here we compared the focused macro-actions against an equal number of "random" macro-

| | | Generated States | Remaining Errors ($\#g$) | Solve Rate |
|---|---|---|---|---|
| 15-Puz. | Primitives only | 30840.5 | 0.0 | 1.0 |
| | Random macros | 72542.3 | 0.0 | 1.0 |
| | Focused macros | **4952.4** | 0.0 | 1.0 |
| Rubik's | Primitives only | >2M | 11.8 | 0.0 |
| | Random macros | >2M | 16.4 | 0.0 |
| | Focused macros | **171331.4** | **0.0** | **1.0** |
| | Expert macros | 30229.1 | 0.0 | 1.0 |

Table 3: Planning results for 15-puzzle and Rubik's cube comparing different action spaces. Random macros perform significantly worse than both primitive actions and focused macros. Trials with macros also include the primitive actions.

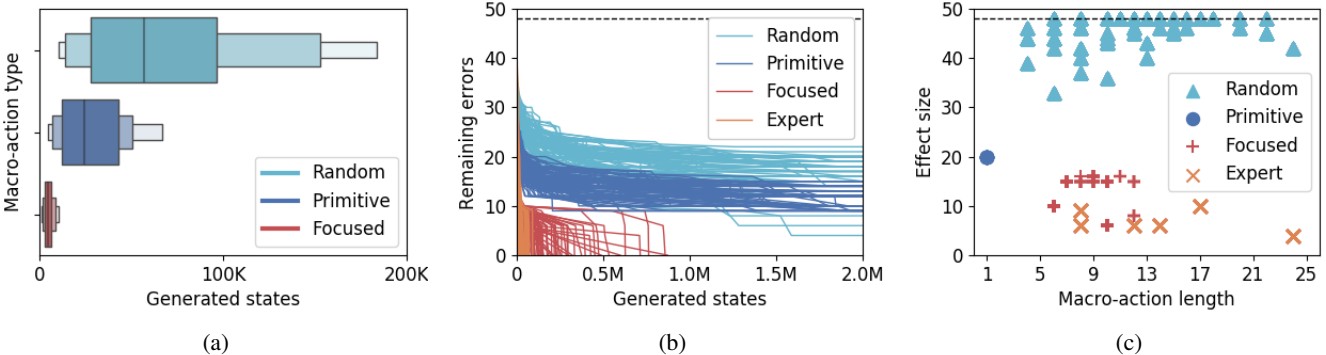

(a)                     (b)                     (c)

Figure 2: (a) 15-puzzle planning efficiency by macro type. Adding focused macros leads to a significant performance improvement over primitive-actions alone. Random macros have the opposite effect. (b) Rubik's cube planning performance by macro type. The vertical axis represents the best observed goal-count value for the number of generated states on the horizontal axis. (c) Effect size vs. length of Rubik's cube macro-actions, by type. (Some points overlap.)

actions of the same length, which were generated (for each random seed) by selecting actions uniformly at random from the valid actions at each state.

We present the results in Table 3, as well as Figures 2a and 2b, where we observe that random macros perform significantly worse than both the primitive actions and the learned focused macros. In both domains, random macros also consistently had larger effect sizes than focused macros. Figure 2c shows a visualization of Rubik's cube macro effect size versus macro length. We suspect the higher planning cost of random macros is partly due to their increased effect size.

## 5.4 Examining Expert Macros in Rubik's Cube

Expert human "speedcubers" use macro-actions to help them manage the Rubik's cube's highly non-focused actions. In speedcubing, the goal is to solve the cube as quickly as possible, without necessarily finding an optimal plan. Most speedcubers learn a collection of macro-actions (called "algorithms" in Rubik's cube parlance) and then employ a strategy for sequencing those macro-actions to solve the cube. Expert macro-actions tend to affect only a small number of state variables, and proper sequencing enables speedcubers to preserve previously-solved parts of the cube while solving the remainder. Common solution methods typically involve multiple levels of hierarchical subgoals and produce plans approximately twice as long as optimal.

As a benchmark, we consider a simplification of the most common expert strategy, where macros are composed of just primitive actions. We select a set of six hand-coded, expert macro-actions to perform various complementary types of permutations.[6] We visualize one of these macro-actions, which swaps three corner pieces, in Figure 3a. Since our simulator uses a fixed cube orientation, we consider all 96 possible variations of each macro (to account for orientation, mirror-flips, and inverses), resulting in 576 total macros—the same number used for the random macro and focused macro trials.

---

[6] Macro-action sequences are included in Appendix D.

In Figure 2c, we plot the effect size and length of each macro, labeled by macro type. We can see that the focused macros have significantly smaller effect size than primitive actions or random macros, and begin to approach the effect size of the expert macro-actions. We also note that the focused macros are somewhat shorter on average than the expert macros, and we suspect that increasing the search budget would result in learning macros with even smaller effects.

In Table 3 and Figure 2b, we compare planning with the expert macros against the other macro types and see that while planning with the expert macros is the most efficient, the learned, focused macros are not far behind. By contrast, the random macros and primitive actions never solved the problem within the simulation budget. We also found that the average solution length for focused and expert macro-actions was about an order of magnitude longer than typical human speedsolve solutions (378 and 319 primitive actions, respectively, vs. ~60 (Speedsolving Wiki 2021)), which suggests that there are additional insights to be mined from human strategy beyond just learning focused macro-actions.

## 5.5 Interpretability of Focused Macros

We examined the learned focused macros for several domains and found that in addition to having low effect size, they were also frequently easy to interpret. In 15-Puzzle, one type of macro swapped the blank space with a central tile; another type exchanged three tiles without moving the blank space. In Rubik's cube, one macro (Figure 3b) swapped three edge-corner pairs while keeping them connected. In Tower of Hanoi, macros moved stacks of disks at a time from one peg to another. We remark that this is quite similar to the

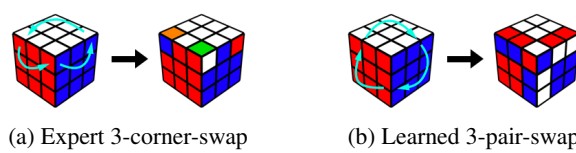

(a) Expert 3-corner-swap      (b) Learned 3-pair-swap

Figure 3: Expert and learned macro-actions (Rubik's cube).

interpretability of the human expert macros in Rubik's cube.

## 5.6 Generalizing to Novel Goal States

Since our macro-generation step is goal-independent, we can reuse previously learned macros to solve problems with novel goal states. To demonstrate this, we generate 100 random goal states for 15-puzzle and Rubik's cube and then solve the puzzles again. In both domains, we find that planning time and solve rate remain effectively unchanged for novel goal states (see Table 4).

# 6 Related Work

The concept of building macro-actions to improve planning efficiency is not new. Dawson and Siklossy [1977] considered two-action macros and analyzed domain structure to remove macros that were invalid or that had no effect. Korf's [1985] Macro Problem Solver investigated how to learn macros in problems with decomposable operators and serializable subgoals. Macro-FF (Botea et al. 2005) and MUM (Chrpa, Vallati, and McCluskey 2014) learned macros from training problem instances and later used them to improve the planning efficiency of testing instances. The macros we discover in Sections 4 and 5 can similarly be reused across problem instances; however, our macro discovery procedure requires neither goal information nor training instances. Our method is perhaps most similar to MARVIN (Coles and Smith 2007), which used macros to escape plateaus during heuristic search, and CAP (Asai and Fukunaga 2015), which decomposed planning problems into subgoals and then found macros to achieve those subgoals. In all of the prior approaches, the learned macros were found to be beneficial for planning, but they also required an explicit model of the domain. Our method is more general than these methods, as it is designed to handle the unique challenges of black-box planning without an explicit model.

Lipovetzky and Geffner [2012] introduced Iterated Width (IW) search, a "blind" planner compatible with black-box simulators, and Lipovetzky, Ramirez, and Geffner [2015] subsequently applied it to planning in Atari video game simulators without known goal states. This work led to the goal-informed Best-First Width Search (BFWS) (Lipovetzky and Geffner 2017; Francès et al. 2017), which we include in our experimental evaluation. Jinnai and Fukunaga [2017] formalized black-box planning and described a method for pruning primitive actions and short macros to avoid generating duplicate states; however their approach did not incorporate goal information.

Recent work by Agostinelli et al. [2019] investigated how to train black-box planning heuristics with neural networks and dynamic programming by strategically resetting the simulator to states near the goal state. Their approach learned heuristics for several domains, including 15-puzzle and Rubik's cube, that supported fast, near-optimal planning. However, training their neural network requires more than 1000 times the simulation budget of our approach, and results in a heuristic that is only informative for a single goal state, whereas ours works for arbitrary goal states.

| Domain | Goal Type | Generated States | Solve Rate |
|--------|-----------|------------------|------------|
| 15-puzzle | Default | 4952.4 | 1.0 |
|  | Random | 4780.0 | 1.0 |
| Rubik's cube | Default | 171331.4 | 1.0 |
|  | Random | 152503.7 | 1.0 |

Table 4: Average planning efficiency and solve rate, when reusing previously-discovered focused macros to solve 15-puzzle and Rubik's cube with either the default goal state or new randomly-generated goal states. Novel goal state performance is effectively unchanged.

# 7 Discussion and Conclusion

We have described a method of learning focused macro-actions that enables reliable and efficient black-box planning across a variety of classical planning domains. While our approach is designed to match the assumptions of the goal-count heuristic, we find that it also improves the performance of more sophisticated black-box planners. Moreover, our method is even competitive with a state-of-the-art LAMA planner, despite the latter having access to a declarative description of the problem.

We are encouraged to see that many of the learned macro-actions had intuitive, interpretable meaning in the task domain. This suggests that our method may be useful for improving explainability in addition to planning efficiency.

This work employed a two-level hierarchy where macro-actions are composed of primitive actions. One extension to bring this method more in line with human-expert techniques would be incorporating additional levels of action hierarchy (i.e. macros composed of other macros), or macros that permit side-effects to certain unsolved variables, combined with macros to subsequently solve those remaining variables. We leave an exploration of these ideas for future work.

## Acknowledgements

We thank Yuu Jinnai and Eli Zucker for many insightful discussions, Tom Silver and Rohan Chitnis for their help with PDDLGym, the anonymous IJCAI'20, AAAI'21, IJCAI'21, and ICAPS'21 HSDIP reviewers for their excellent feedback on earlier versions of this paper, and our colleagues at IBM Research and Brown University for their thoughtful conversations and support. This research was supported in part by the ONR under the PERISCOPE MURI Contract N00014-17-1-2699.

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

# Appendix

## A   Suitcase Lock Implementation

Each Suitcase Lock problem instance has $N$ dials, each with $M$ digits, and $2N$ actions, half which increment a deterministic subset of the dials (modulo $M$), and half which decrement the same dials (see Figure 1). Let $k_i$ denote the effect size of action $a_i$, and $\bar{k}$ denote the mean effect size across all actions. Given a $\bar{k}$, we generate problem instances with different start states, goal states, and sets of actions such that they have mean effect size $\bar{k}$.

   We always ensure that every state can be reached from every other state. Note that if $\bar{k} = N$, or if all actions modify (for example) an even number of state variables, it is not possible to reach every state from every other state. To circumvent this issue, we check that for a given problem instance, the increment and decrement action sets can each be reduced to an $N \times N$ binary matrix with full rank. We repeatedly generate action sets with the desired mean effect size until we find one that satisfies this condition. The resulting action sets are therefore different for each random seed, except when $\bar{k} = 1$ where we always use the identity matrix $I$, and when $\bar{k} = (N - 1)$ where we use $1 - I$ with an extra 1 added to the first diagonal element to break symmetry. The decrement actions are always the negation of the increment actions, and we ignore them for $M = 2$.

# B  Simulator Details

## B.1  PDDLGym

We use the PDDLGym library (Silver and Chitnis 2020) to automatically construct black-box simulators for PDDL planning problems. State information is represented as a variable-length list of currently-true literals. The planning agent has access to this state information, along with the goal (represented as a conjunction of literals), the action applicability function, and the simulator function. We chose a representative set of PDDL problems and executed uniform random actions to generate 100 unique random starting states for each, keeping the goal fixed. The associated `.pddl` files can be found in the code repository.

## B.2  15-Puzzle

The 15-puzzle is a $4 \times 4$ grid of 15 numbered, sliding tiles and one blank space (see Figure 4a). The puzzle begins in a scrambled configuration, and the objective is to slide the tiles until the numbers are arranged in increasing order. There are approximately $10^{13}$ states and the worst-case shortest solution requires 80 actions (Brüngger et al. 1999). Our simulator uses a state representation with 16 variables (for the positions of each tile and of the blank space), and 48 primitive actions (that swap the blank space with one of the adjacent tiles), of which only 2–4 can be applied in each state. Similarly, macro-actions can only run if they begin with the correct blank space location.

We set the macro-learning budget $B_M = 32,000$ simulator queries, the number of macros $N_M = 192$, and the number of repetitions $R_M = 16$. The budget was chosen to approximately match the number of steps required to solve one problem instance with primitive actions. This resulted in 12 generated macros per repetition, and a per-repetition simulator budget of 2000 state transitions. We compared these macro-actions against 192 "random" macro-actions of the same lengths, which were generated (for each random seed) by selecting actions uniformly at random from the valid actions at each state.

We then solved the 15-puzzle using greedy best-first search with the goal-count heuristic and a simulation budget of $B_S = 500,000$ state transitions. We generate 100 unique starting states by scrambling the 15-puzzle with uniform random actions for either 225 or 226 steps, with equal probability (to ensure that we see all possible blank space locations). The resulting puzzles can be found in the code repository.

**Note B.2.1.** *On Finding States Where Macro Preconditions Do Not Apply*

As mentioned in Section 4, the macro-learning procedure includes an option to repeat the search $R_M$ times from new starting states where the previously-discovered macros are not applicable. In general, finding such a state can be as hard as planning, although it might be easier if there is no requirement for generating a plan, e.g. by resetting the simulator to generate a new starting state. Some environment implementations do not allow resetting to arbitrary states, and, in those cases, a plan must be generated. 15-puzzle is the only domain where we use $R_M > 1$, and for this domain, we found that either state generation strategy (random walk or simulator resets) was effective. For domains where the simulator cannot be reset, and where a random walk is insufficient, it is possible to make the search more informed, such as by incorporating state novelty into the heuristic.

## B.3  Rubik's Cube

The Rubik's cube is a $3 \times 3 \times 3$ cube with colored stickers on each outward-facing square (see Figure 4b). The puzzle begins in a scrambled configuration, and the objective is to rotate the faces of the cube until all stickers on each face are the same color. There are approximately $4.3 \times 10^{19}$ states, and the worst-case shortest solution requires 26 actions (Rokicki 2014). Our simulator fixes a canonical orientation of the cube, and uses a 48-state-variable representation (for the positions of each colored square, excluding the stationary center squares). The problem has 12 primitive actions (i.e. rotating each of the 6 faces by a quarter-turn in either direction), and these actions are highly non-focused: each modifies 20 of the 48 state variables.

We set the number of learned macro-actions $N_M = 576$ so that we could fairly compare the generated macro-actions against our set of expert macro-actions. We learned macro-actions from a single starting state $R_M = 1$, and set a simulation budget of $B_M = 1,000,000$ simulator queries. We also compared against 576 "random" macro-actions of the same lengths as the expert macros (six distinct macro-actions plus their corresponding variations), which were regenerated for each random seed. We set the search budget $B_S = 2,000,000$ simulator queries.

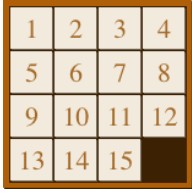

(a) 15-Puzzle

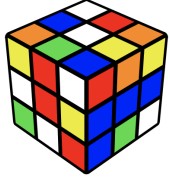

(b) Rubik's Cube

Figure 4: Visualizations of the planning domains that use domain-specific simulators

We obtained starting states for Rubik's cube from modified versions of the 100 hardest problems from Büchner [2018]. The problems were specified as random sequences of primitive actions to be applied to a solved Rubik's cube in order to generate the starting state, as well as a corresponding $SAS^+$ representation for each problem. The original Büchner problems incorporated 18 half-turn and quarter-turn action primitives, whereas our simulator uses only 12 quarter-turn action primitives. Our modification removed the 6 half-turn actions from the $SAS^+$ representation and converted problem specifications involving half-turns to their equivalent quarter-turn-only specifications. The resulting problems consisted of between 12 and 29 primitive actions, with an average of about 20. (We also tried generating starting states by scrambling the cube with uniform random actions for 60 steps, with similar results.) The problems we use, and the procedure we use to generate randomly scrambled starting states, can be found in the linked code repository.

# C    Updating the Simulator with Macros

## PDDLGym

For the PDDLGym simulators, we build new macro-operators for the saved primitive-action sequences by:

1. Re-binding the original lifted parameters to new variables that capture any dependencies between subsequent actions. For example, the sequence [PLACE_ON(B,C), PLACE_ON(A,B)], would result in two distinct parameters for objects A and C, plus a third, shared parameter for object B that is reused by both primitives.

2. Combining the preconditions of subsequent primitive actions when they are not already met by the effects of previous primitive actions. For example, if ACTION1 has precondition (A and B) and effect C, and ACTION2 has precondition (C and D), this would result in the combined precondition (A and B and D).

3. Combining and simplifying the effects of the primitive actions to remove unnecessary negations. For example, if the combined precondition so far is A, and if ACTION1 has effect (B and (not A)) and ACTION2 has effect (C and A), this would result in a combined effect of (B and C), since A is already a precondition.

We present pseudocode in Algorithm 2, and the implementation and the resulting macro-augmented PDDL files can be found in the code repository.

Note that while the desired number of macro-actions for all PDDLGym domains was set to $N_M = 8$, we were only able to find four unique macros for the doors domain.

---

**Algorithm 2** Construct lifted macro for PDDLGym

---

**Input**:
*actions*, a sequence of grounded primitive actions
*operators*, map from names to lifted primitive operators
**Output**:
*macro*, a newly-constructed, lifted macro-operator

1:  *macro.params* := ∅
2:  *macro.preconds* := ∅
3:  *macro.effects* := ∅
4:  *lifted* := map from grounded to lifted variable names
5:  **for** *action* in *actions* **do**
6:      *op* := *operators*[*action.name*]
7:      *lifted*.update({ *v* ↦ new_variable_name(), for *v* in *action.variables* if *v* not in *lifted*})
8:      *binding* := {*p* ↦ *lifted*[*v*], for (*p*, *v*) in zip(*op.params*, *action.variables*)}
9:      **for** *p* in *op.params* **do**
10:         *macro.params*.add( *binding*[*p*] )
11:     **end for**
12:     **for** *literal* in bind_literals(*op.preconds*, *binding*) **do**
13:         **if** *literal* not in *macro.effects* and *literal* not in *macro.preconds* **then**
14:             *macro.preconds*.add(*literal*)
15:         **end if**
16:     **end for**
17:     cleanup_contradictory_effects(*op.effects*)
        // Simplify any contradictory effects to just their positive part, e.g. ((not A) and A) becomes (A)
18:     **for** *literal* in bind_literals(*op.effects*, *binding*) **do**
19:         **if** (¬*literal*) in *macro.effects* **then**
20:             *macro.effects*.remove(¬*literal*)
21:         **else**
22:             *macro.effects*.add(*literal*)
23:         **end if**
24:     **end for**
25: **end for**
26: **return**  *macro*

---

## Domain-Specific Simulators

For 15-puzzle and Rubik's cube, both simulators use a position-based representation (i.e. the positions of each numbered tile or blank space; the positions of each colored sticker excluding the stationary center stickers). Primitive actions are expressed as permutations operations on the indices of the state variables.

To augment the simulator with macro-actions, we computed the overall permutation for each sequence of primitive actions, and store the result (along with its precondition, if any) as a new permutation operation that the simulator can apply using the same procedure it uses for primitive actions.

In the case of Rubik's cube, none of the primitive actions have preconditions, so the resulting macros do not have preconditions either. However, for 15-puzzle, primitive-action preconditions depend on the position of the blank space. Fortunately, since we only construct macros for valid action sequences and since actions deterministically modify the position of the blank space, as long as the initial precondition is satisfied, each action will automatically satisfy the precondition of the next action in the sequence. Thus, when saving each 15-puzzle macro-action, we simply keep track of the blank-space location required to execute its first primitive action, along with its overall permutation.

The code to generate the overall permutation of a 15-puzzle or Rubik's cube macro-action can be found in the corresponding module in the code repository.

# D  Expert Rubik's Cube Macros

We use the following expert macro-actions (expressed in standard cube notation (Singmaster 1981)):

- 3-corner swap (see Figure 3a): $L'\ B\ L\ F'\ L'\ B'\ L\ F$
- 3-edge swap, middle: $L'\ R\ U\ U\ R'\ L\ F\ F$
- 3-edge swap, face: $R\ R\ U\ R\ U\ R'\ U'\ R'\ U'\ R'\ U\ R'$
- 2-corner rotate: $R\ B'\ R'\ U'\ B'\ U\ F\ U'\ B\ U\ R\ B\ R'\ F'$
- R-permutation: $F\ F\ R'\ F'\ U'\ F'\ U\ F\ R\ F'\ U\ U\ F\ U\ U\ F'\ U'$
- 2-edge flip: $L\ R'\ F\ L\ R'\ D\ L\ R'\ B\ L\ R'\ U\ U\ L\ R'\ F\ L\ R'\ D\ L\ R'\ B\ L\ R'$

To generate the full set of 576 expert macro-actions, we consider 96 variations of each of the above sequences, including all 24 possible orientations, along with their inverse and mirror-flipped versions.

The learned 3-pair-swap macro in Figure 3b was not included with the expert macro-actions. We also provide its action sequence for completeness.

- 3-pair-swap (see Figure 3b): $F'\ L\ F'\ L'\ F\ F\ R\ U'\ R'\ F'\ U\ F$

# E   Reproducibility

## E.1   Hyperparameter Selection

In the paper, and the preceding sections of the appendix, we describe the final hyperparameters used to run the experiments. We arrived at these values by an informal hyperparameter search, and many hyperparameters never changed from their initial values.

**PDDLGym Domains**   For the PDDLGym domains, the simulation budget was set to 100K queries for compute reasons, as the PDDLGym simulator was slower than the domain-specific simulators. The macro-learning budgets for the PDDLGym domains were set to be comparable to the number of simulator queries needed to solve a single problem instance using greedy best-first search with the goal-count heuristic and primitive actions. The number of PDDLGym macros was chosen to be uniform across the various domains.

We ran some informal experiments with different amounts of macros to ensure that the approach was not overly sensitive to the number of macros, and found that there was no significant change in performance when adding more macros, as long as the effect size remained low. We found that it was possible to tune the number of macros for each PDDLGym domain separately, with improved results, but felt that leaving the number of macros fixed was a more principled evaluation of our approach.

**15-Puzzle**   The simulation budget for 15-puzzle was set to 500K queries, although this full simulation budget was not needed since every problem was solved in fewer than that many generated states. The number of macros for 15-puzzle was set higher than for PDDLGym, to compensate for the fact that the domain-specific simulator macros are tied to specific tiles, rather than lifted like the PDDLGym macros. The numbers of random and focused macros were equal to each other, to ensure a fair comparison.

**Rubik's Cube**   We increased the Rubik's cube simulation budget to 2M queries to see whether the primitive-action planner could solve any problems with more planning time. The macro-learning budget for Rubik's cube was set to 1M queries to see if the total cost of learning macros and planning was low enough to justify learning macros for a single problem instance. The numbers of focused and random macros for Rubik's cube were chosen to match the number of expert macro-actions, which was itself chosen so that the expert macros could efficiently solve the Rubik's cube.

## E.2   Computational Resources

This paper included experiments that ran a cluster of Linux machines running either RedHat 7.7 or Debian 10, with varying hardware specifications. However, a single seed for each of the experiments can run in 30 minutes (and usually significantly less) on a MacBook Pro running macOS Mojave (10.14.6), with 2GHz i5 processor and 16GB RAM. No GPUs were used for any of the experiments.

## E.3   Random Seeds

Random seeds were used to generate the problem instances, macro-actions, and planning results. We have attempted to make results as reproducible as possible by fixing random seeds. The commands listed in the *README* file should reproduce our results exactly. As noted in the preceding sections of this appendix, we have also saved and included the generated problem instances in the linked code repository, to allow for maximum portability.