# OpenReview forum: "Efficient Black-Box Planning Using Macro-Actions with Focused Effects"
_icaps-conference.org/ICAPS/2021/Workshop/HSDIP — HSDIP 2021_

### Official Review · AnonReviewer1 · 2021-05-25

**Confidence:** 4
**Overall Score:** Accept

**Review:**

The paper introduces a way macros with a small number of effects can be created for blackbox planning. The hypothesis is that macros with fewer effects will lead to better correlation of the goal-count heuristic with the true goal distances and thus improve planning performance.

A large part of the paper consists of experiments that confirm this hypothesis and investigate a method to find such macros by a best-first search. The search uses the length of the macro and the size of its effect as a cost and starts from a random state where previous macros are not applicable. It is unclear to me how such a state can be found in a blackbox planning framework other than repeatedly random walking from known states. If that is how these states are found, I'd be interested in the effort required for this (especially if many macros should be created, finding a state in which no macro is applicable sounds challenging). The other part of the search requires finding the size of the macro's effect. Without access to the model, all the blackbox search can do is to observe the effect on the state where the macro was applied. I view this as slightly problematic because the actions could have additional effects that are not visible in the particular state where the macro is applied (for example, because an action sets a variable to true that already is true).

Anyway, the experiments show a clear benefit of the discovered macros with a greedy search as well as one that is based on novelty. The experimental design is well done, with the exception that the number of macros and the budget allocated to find them is hand-tuned and varies from domain to domain quite drastically. The experiments thus show that there is potential in using such macros if these parameters are picked correctly but give no indication how to pick them. It also seems that there is no parameter setting that would perform well in all domains as the chosen parameters are far apart.

Despite this, the paper is a solid contribution and should be accepted.

---

> ### Author Response · Authors · 2021-06-01
> **Response from Authors**
>
> Thank you for your comments.
>
> Your point about finding states where the previous macros are not applicable is well taken. In practice, we only used this to find additional macros in 15-puzzle, and in that domain it is quite easy to find such states by random walk. However, it is possible, even in the black-box setting, to make the search more informed, for example by incorporating some sort of novelty-based heuristic.
>
> Regarding actions that potentially have additional, unobservable effects: one way to deal with this would be to treat that behavior as a conditional effect. In the example you give where the action sets a variable to 'true' in a state where it is already true, our effect size definition simply ignores that variable in the effect, whereas in some other state, the variable might be included. Technically, we would need to evaluate each macro from every valid state to determine its effect size, which is clearly infeasible. In practice, we simply measure each macro's effect size once, and assume it doesn't change (although we could easily relax this assumption by running the macro from multiple states).
>
> The number of macros for 15-puzzle was set higher to compensate for the fact that the domain-specific simulator macros are tied to specific tiles, rather than lifted like the PDDLGym macros. We have since run an additional experiment on 15-puzzle which brought the number of macro-actions down from 1600 to 192, and this did not substantially change the results (now 4952.4 generated states for GBFS instead of 3980.4 previously; both substantially less than 30K with primitive actions). In Rubik's cube, we had the additional constraint of matching the number of expert macros that were needed to reliably solve every puzzle.

---

> > ### Comment · AnonReviewer1 · 2021-06-02
> > **Thanks, please add those answers to the paper**
> >
> > Thank you for your response.
> >
> > Could you please add the explanation how to find states where macros are not applicable to the paper as well? It wasn't clear to me that this was done by a random walk. The main problem I see with this is that the negation of all macro preconditions is a new goal, so finding such a state should (in general) be as hard as finding a plan for the original problem. It could even be harder, as the goal no longer is a conjunction of literals.
> >
> > Your answer about how to determine the effect size would also make a good addition to the paper.
> >
> > The discussion of how the budget and number of macros were chosen could be added to the paper as well. I still see it as a downside of the method that these numbers have to be selected manually. However, finding good choices automatically could easily be deferred to future work and I don't think it is necessary to solve this as part of this submission. Discussing the issue and potential solutions would improve the paper, though. The results you discussed in your response are encouraging and show that the method is somewhat robust wrt this parameter but it doesn't completely solve the issue.

---

> > > ### Author Response · Authors · 2021-06-02
> > > **Follow-up from Authors**
> > >
> > > We would be happy to include an explanation of the process for finding such states.
> > >
> > > You are correct that finding a *plan* to such a state should in theory be as hard as planning, but finding a state might be easier if there is no requirement for generating a plan, e.g. by resetting the simulator to generate a new starting state. Some environment implementations do not allow resetting to arbitrary states, and, in those cases, a plan must be generated. Luckily, in practice, it is often quite easy to reach a state where the precondition does not hold and this can be done quite effectively even with a random walk.

---

> > > > ### Comment · AnonReviewer1 · 2021-06-02
> > > > **Finding states**
> > > >
> > > > I see, I wasn't aware that resetting to a specific state was an option as well. In some domains constructing a state from scratch could easily produce a state that doesn't make sense (for example a blocksworld instance where a block is stacked on itself). Anyway, this second option is just another reason to discuss this process in the paper. Thanks for adding it.

---

### Official Review · AnonReviewer2 · 2021-05-26

**Confidence:** 4
**Overall Score:** Accept

**Review:**

**Tittle: Efficient Black-Box Planning Using Macro-Actions with Focused Effects**

### Summary
In black-box planning the state and goal descriptions are available, but no information about the action model. Thus, techniques to guide the search are quite uninformed. Often goal counting is used. The authors observe that the more state variables are changed  by an action, the less informed goal counting is and the longer the search takes. Thus, they propose to learn macro-actions (a sequence of actions) which modify as few variables as possible.
The authors evaluate on several domains (with 100 instances per domain) the performance of GBFS and BFWS using the given actions or using additionally the macro actions. Using their macro actions reduced the number of generated states drastically.
Furthermore, their macros where compared to humanly crafted expert macros and to random macros. Their macros are better than random macros, the handcrafted expert macros are even smaller and more focused than their automatically derived macros.

### Feedback
Thank you for the interesting submission. The paper is a wonderful fit for HSDIP, but requires additional information. The technique you want to present is insufficiently explained. The most important part of your algorithm is HOW candidate macros are generated. After reading section 4 and appendix B I still do not know how this is done. You write that you start a best-first search with a simulation budget of $B_M$, but I have no clue on which state space you do the best-first search, how you extract the macros, and how you use the simulation budget. My guess is that you just start a search **without caring for a goal** from the sampled state. Due to the heuristic, the search will select some actions. For every generated (or expanded?) state, a macro which consists of the actions used to reach this state is added to the candidate set. The algorithm stops after $B_M$ states have been generated (or expanded). Some information like this (this is just how I guess you did it) has to be added.

In you experiments you wrote that you generate 100 instances per domain with unique states and a fixed goal condition. If you tell me that you generate 100 instances of a domain, I assume by default that those instances describe different state spaces. This is in contrast to the information of the "fixed goal". For example in Towers of Hanoi, the goals depends on the number of disks and cannot be fixed unless the number of disks is fixed. Furthermore, I expected to learn how you generalize the action macros from state space to another state space. After taking a look at the submitted PDDL files, I realized the state spaces of all instances of a domain are the same. Now everything makes sense. Please state explicitly in the abstract and methodology section that you generate macro actions for a single state space. This prevents the reader from wrong expectations.

I really like your project idea and think this will be a great paper, once the macro generation is cleared up.

### Questions
- Please explain to me again how the macros are generated.
- Do you have an idea how you can generalize macro actions within a domain (given the black box planning)?
- Do you see an application for macro action in regular planning (when the model is given)?

### Minor Issues
- *Figure 1:* When referring to figure 1, add also the information whether you refer to figure 1 top or bottom.
- The references change arbitrarily between abbreviating and not abbreviating some conferences/journals. Especially, the same conference/journal is sometimes abbreviated and sometimes not. And especially the reference Frances et al. 2017 contains a lot of information the others do not.

---

> ### Author Response · Authors · 2021-06-01
> **Response from Authors**
>
> Thank you for the feedback.
>
> The macro search procedure is exactly as you describe. It is goal-agnostic, and, as heuristic search generates new states, the action sequences associated with those states are added to a priority queue where priority is based on effect size. As a result, when the queue becomes full, the action sequences with largest effect size will be evicted first.
>
> Consider an example with two primitive actions a1 and a2, where BFS starts at state s0. Expanding s0, the action a1 generates state s1, and a2 generates s2. Expanding s1, a1 generates s3 and a2 generates s4. Thus macro m4, corresponding to state s4, would be the action sequence [a1, a2], and we would evaluate its net effect by comparing s4 with s0. We had included this small example in Appendix B, along with the full algorithm for the macro-learning procedure, but in light of your comments, we think it would be clearer to move both to the the main body of the paper.
>
> You are correct that the state space is fixed for all problem instances in each domain. We feel that this better aligns with the conventions of black-box planning, but we are happy to include additional clarification. Note that in Rubik's cube, for example, each puzzle instance is simply a different state within the same state space, and it is unclear what it would even mean to change the state space.
>
> We show in Section 5 that the same set of macro actions is beneficial for different start states as well as different goal states. As for generalizing to different state spaces, some care would need to be taken to ensure that the macros still apply, but in principle this should be possible. For example, in Towers of Hanoi, the macros we learn would easily generalize to a problem with *more* disks (though perhaps not fewer), since they are already lifted. We describe the process by which we construct these lifted macro-operators in Appendix E. (Note that in the black box setting, care must also be taken to ensure that the *primitive* actions still apply in new instances.)
>
> We suspect that our approach would also be beneficial when the planner has access to the primitive action models. We plan to investigate this idea more in future work.

---

### Decision · Program_Chairs · 2021-06-10

**Decision:**

Accept

**Comment:**

The reviewers agreed that the paper should be accepted and both asked for additional clarification on the details of the method. We ask the authors to use the information in the reviews and the discussion to improve this aspect of the paper.